# Physics-Informed Neural Networks with Message-Passing Weights

## Abstract

Adaptive loss balancing algorithms play a crucial role in improving the performance of Physics-Informed Neural Networks (PINNs) by effectively managing the weights assigned to different loss components. Most notably, Wang et al. (2022a) introduced Causal Physics-Informed Neural Networks (Causal PINNs), which achieve superior performance by simply reformulating the loss function based on the causal structure that emerges from time dependency. However, despite their empirical success, a solid theoretical analysis for the effectiveness of Causal PINNs has not received adequate attention. This paper addresses this gap by providing a theoretical rationale for Causal PINNs through the Belief Propagation (BP) algorithm, which is commonly used for causal inference. In addition, motivated by this analysis, we propose a Message Passing PINNs (MP-PINNs), a novel adaptive weighting algorithm. Through extensive numerical experiments, we demonstrate that the proposed MP-PINNs significantly outperform existing adaptive weighting methods, exhibiting superior performance in solving complex PDEs. Our findings highlight the potential of MP-PINNs as a powerful tool to enhance both the accuracy and efficiency of PINNs.

## 1 Introduction

Physics-Informed Neural Networks (PINNs) have emerged as a powerful framework for solving scientific and engineering problems by integrating physical laws directly into the learning process. These networks leverage the principles of physics to guide the training of neural networks, allowing them to learn not only from data but also from the underlying governing equations of the phenomena being modeled. This innovative approach has been successfully applied to various domains, including fluid dynamics, structural mechanics, and heat transfer, demonstrating significant improvements in accuracy and generalization capabilities compared to traditional data-driven methods Raissi et al. (2019); Karniadakis et al. (2021).

Adaptive loss balancing algorithms are essential for enhancing the performance of PINNs by managing the weights assigned to different loss components . These algorithms help mitigate potential imbalances that may arise during the training process, thereby improving the overall efficiency of data-driven learning that incorporates physical principles. PINNs have demonstrated considerable promise in addressing a wide range of scientific and engineering challenges, making them valuable tools in these fields.

Causal Physics-Informed Neural Networks (Causal PINNs) represent a significant advancement in this area, achieving superior performance by explicitly reformulating the loss function to respect causality Wang et al. (2022a). However, despite their empirical success, a solid theoretical justification for the effectiveness of Causal PINNs has been lacking. The absence of a robust theoretical foundation may limit the applicability and extension of these approaches, posing a critical barrier to the advancement of PINNs.

The Belief Propagation (BP) algorithm is a message-passing (MP) algorithm used for causal inference. For instance, in certain scientific fields, if causality is established between instances, BP can be utilized to infer their exact causal relationships Chang et al. (2014). Also the mechanism of the MP algorithm is already widely applied in machine learning and deep learning areas (Scarselli et al., 2008; Gilmer et al., 2017; 2020). However, there have been no attempts to connect this to adaptive

weighting algorithms for PINNs. We note that Causal PINNs only consider time-dependency, although spatial dependency may also influence the calculate of the residual loss. Furthermore, Daw et al. (2022) discussed how information from the initial and boundary conditions can propagate to interior collocation points. In this context, the BP algorithm may suggests a method for analyzing the causal relationships between each collocation point.

This paper addresses the gap between the performance of Causal PINNs and the lack of analytical understanding by providing a theoretical reformulation of their algorithms based on the BP algorithm. Furthermore, building on this analysis, we propose a novel adaptive weighting algorithm, termed Message Passing PINNs (MP-PINNs). Through extensive numerical experiments, we demonstrate that the proposed MP-PINNs significantly outperform existing adaptive weighting methods, exhibiting superior performance in solving complex partial differential equations (PDEs). Our findings highlight the potential of MP-PINNs as a powerful tool for enhancing both the accuracy and efficiency of PINNs.

The main contributions of this work can be summarized as follows:

- We reframe Causal PINNs from the perspective of the BP algorithm

- Inspired by this analysis, we propose a novel adaptive weighting algorithm termed Message-Passing PINNs (MP-PINNs)

- We demonstrate, through numerical experiments, that the proposed MP-PINNs significantly outperform existing adaptive weighting algorithms.

## 1.1 RELATED WORKS

**Failure modes of PINNs** Despite the applicability of PINNs for solving various PDEs, there are still various problems to solve. Escepcially, (Krishnapriyan et al., 2021) presented a possible failure mode of PINN, describing the challenging loss landscape that can appear in time-dependent PDEs.

**Adaptive weights of PINNs** One strategy to improve PINN training is the use of adaptive weighting methods. For example, McClenny & Braga-Neto (2020) introduced a self-adaptive approach based on a soft attention mechanism, where weights are trained adversarially. Son et al. (2023) framed PINNs as a constrained optimization problem and applied the augmented Lagrangian method to define an unconstrained minimax problem. By solving this using gradient descent and ascent, they demonstrated the effectiveness of a loss-balancing algorithm. Additionally, Wang et al. (2020) proposed a balancing method that directly influences gradient statistics, while Wang et al. (2022b) explored the Neural Tangent Kernel of PINNs to further enhance training.

**Causal PINNs** To address the failure modes of PINNs, (Wang et al., 2022a; Penwarden et al., 2023) defined this issue as the failure to deliver accurate information to collocation points that are distant from the initial points. Furthermore, they employed weighted loss to train PINNs, using namely causal weights. However this approach only considers the causality of time dependency rather than spatial dependency.

## 2 BACKGROUND

### 2.1 CAUSAL WEIGHTS FOR TRAINING PINNS

The causal PINN is based on the *causal weights* which is defined on the time-dependent PDEs (Wang et al., 2022a). For details, they consider the specific form of PDE given as

$$\mathcal{R}[u](t, x) := \partial_t u(t, x) + \mathcal{N}[u](t, x) = 0, \qquad t \in [0, T), x \in \Omega$$

subject to the initial and boundary conditions

$$\mathcal{I}[u](0, x) = 0, \qquad x \in \Omega,$$
$$\mathcal{B}[u](t, x) = 0, \qquad t \in [0, T), x \in \partial\Omega$$

where $N[\cdot]$ is a linear or nonlinear differential operator, $B[\cdot]$ is a boundary operator and $u$ is a unknown solution. To solve this, PINN loss generally defined as follows:

$$\mathcal{L}(\theta) = \lambda_{ic}\mathcal{L}_{ic}(\theta) + \lambda_{bc}\mathcal{L}_{bc}(\theta) + \mathcal{L}_r(\theta), \tag{1}$$

$$\mathcal{L}_{ic}(\theta) = \frac{1}{N_{ic}}\sum_{i=1}^{N_{ic}}|\mathcal{I}[u_\theta](0, x_{ic}^i)|^2, \tag{2}$$

$$\mathcal{L}_{bc}(\theta) = \frac{1}{N_{bc}}\sum_{i=1}^{N_{bc}}|\mathcal{B}[u_\theta](t_{bc}^i, x_{bc}^i)|^2 \tag{3}$$

and the residual loss $\mathcal{L}_r$ of general equation is defined as the weighted sum over fixed time grids $t_1, \ldots, t_{N_t}$ dividing $[0, T)$, using causal weights given as follows:

$$\mathcal{L}_r(\theta) = \sum_{i=1}^{N_t} w_i(\theta)\mathcal{L}_r(t_i; \theta),$$

$$\mathcal{L}_r(t; \theta) = \frac{1}{N_x}\sum_{j=1}^{N_x}|\mathcal{R}[u_\theta](t, x_j)|^2,$$

$$w_i(\theta) = \exp(-\epsilon\sum_{k=1}^{i-1}\mathcal{L}_r(t_k; \theta))$$

where *causality parameter* $\epsilon$ determines the steepness of the weight $w_i$, which is controlled through an annealing strategy involving an increasing sequence of $\epsilon$ values.

## 2.2 Belief-Propagation algorithm

In this section, we introduce the Belief Propagation (BP) with sum–product message passing which is used to infer a marginal probability for each single random variables over a given Markov random field. More specifically, we only consider the case that every potential function $\psi$ related to each factor of the random field has two input variables, i.e. the joint probability has the form

$$\mathbb{P}(X_i = x_i, \forall i \in V) = \prod_{\{i,j\}\in E}\psi_{i,j}(x_i, x_j).$$

with random variables $X_i : S_i \to \mathbb{R}$ corresponds to each nodes $i$ of given factor graph $G = (V, E, \psi)$. The BP algorithm for marginal inference is consisting of two stages: (1) *message-passing* process $MP$ and (2) *normalization* process $NM$. Through these stages, the *message* $\mu_{i\to j} := \mu_{\{i,j\}\to j} : S_j \to [0, 1]$ from factor $\{i, j\} \in E$ to node $j \in V$ is calculated iteratively, i.e. $\mu^{(\tau+1)} := \{\mu_{i\to j}^{(\tau+1)}\}_{\{i,j\}\in E} = (NM \circ MP)(\mu^{(\tau)})$ for each iteration step $\tau$. First, the massage-passing process is as follows:

$$MP(\mu^{(\tau)})_{i\to j}(x_j) = \sum_{x_i\in S_i}\psi_{i,j}(x_i, x_j)\prod_{k\in N_i\setminus\{j\}}\mu_{k\to i}^{(\tau)}(x_i), \tag{4}$$

for each $\{i, j\} \in E$, $x_j \in S_j$ and iteration $\tau$ where $N_j$ denotes the neighborhood of $j$, i.e. $N_j := \{v \in V : \{j, v\} \in E\}$. Next, for the normalization process, All messages are normalized in L1 sense, i.e.

$$\mu_{i\to j}^{(\tau+1)}(x_j) := NM(MP(\mu^{(\tau)}))_{i\to j} = \frac{MP(\mu^{(\tau)})_{i\to j}(x_j)}{\sum_{x\in S_j}MP(\mu^{(\tau)})_{i\to j}(x)} \text{ for each } x_j \in S_j.$$

Note that when G has no loops, then the BP algorithm naturally requires only a single iteration. However, if there's a loop, then it conditionally converges. After $\tau$ iterations, the marginal inference for $\mathbb{P}(X_i = x_i)$ for each $X_i$ is conducted by calculating *belief* $b_i^{(\tau)} : S_i \to [0, 1]$ for each node $i$ as

$$\mathbb{P}(X_i = x_i) \approx b_i^{(\tau)}(x_i) = \frac{\prod_{j\in N(i)}\mu_{j\to i}^{(\tau)}(x_i)}{\sum_{x\in S_i}\prod_{j\in N(i)}\mu_{j\to i}^{(\tau)}(x)} \text{ for each } x_i \in S_j.$$

## 3 THE CONNECTION BETWEEN THE BP ALGORITHM AND CAUSAL WEIGHTS

In this section, we will demonstrate how the causal weights can be connected to the causal inference over a Markov random field and derived using the BP algorithm. In the formulation of 2.1, the factor graph $G = (V, E, \psi)$ is defined as:

$$V = \{v_i : i = 1, \ldots, N_t,\}$$
$$E = \{e_i := (v_i, v_{i+1}) : i = 1, \ldots, N_t - 1\}$$

based on the collocation points $(t_i, x_j)$ for training PINN $u_\theta(t, x)$ with trainable parameter $\theta$. Furthermore, for each random variable $X_i := X_{v_i} : S_i \to \mathbb{R}$ related to the node $v \in V$, the sample space $S_i$ is defined as $\{-1, 1\}$, and the joint probability of $\{X_i\}_{i=1}^{N_t}$ is defined as

$$\mathbb{P}(X_i = x_i, i = 1, \ldots, N_t; \theta) = \prod_{i=1}^{N_t - 1} \psi_{e_i}(x_i, x_{i+1}; \theta)$$

where the potential function $\psi_{e_i} : S_i \times S_{i+1} \to [0, \infty)$ is defined as

$$\psi_{e_i}(x_i, x_{i+1}; \theta) = \begin{cases} \exp(-\epsilon \mathcal{L}_r(t_i; \theta)), & \text{for } x_i = 1, x_{i+1} = 1, \\ 1 - \exp(-\epsilon \mathcal{L}_r(t_i; \theta)), & \text{for } x_i = 1, x_{i+1} = -1, \\ 0, & \text{for } x_i = -1, x_{i+1} = 1, \\ 1, & \text{for } x_i = -1, x_{i+1} = -1 \end{cases}$$

for each $i = 1, \ldots, N_t - 1$.

To find the marginal $\mathbb{P}(X_i)$ for each $X_i$, the message-passing process of BP algorithm runs as follows:

$$\mu_{i \to i+1}(x_{i+1}) = \begin{cases} 1 \cdot \mu_{i-1 \to i}(-1) + (1 - \exp(-\epsilon \mathcal{L}_r(t_i; \theta))) \cdot \mu_{i-1 \to i}(1), & (x_{i+1} = -1) \\ 0 \cdot \mu_{i-1 \to i}(-1) + \exp(-\epsilon \mathcal{L}_r(t_i; \theta)) \cdot \mu_{i-1 \to i}(1) & (x_{i+1} = 1) \end{cases}$$

for $i = 2, \ldots N_t - 1$ and

$$\mu_{i+1 \to i}(x_i) = \begin{cases} 1 \cdot \mu_{i+2 \to i+1}(-1) + 0 \cdot \mu_{i+2 \to i+1}(1), & (x_i = -1) \\ (1 - \exp(-\epsilon \mathcal{L}_r(t_i; \theta))) \cdot \mu_{i+2 \to i+1}(-1) + \exp(-\epsilon \mathcal{L}_r(t_i; \theta)) \cdot \mu_{i+2 \to i+1}(1) & (x_i = 1) \end{cases}$$

for $i = 1, \ldots N_t - 2$ where the boundary conditions are given as

$$\mu_{1 \to 2}(1) = \exp(-\epsilon \mathcal{L}_r(t_1; \theta)), \mu_{N_t \to N_t - 1}(1) = 0.5.$$

Finally, we obtain the marginal

$$\mathbb{P}(X_i = 1) = b_i(1) = \mu_{i+1 \to i}(1) \cdot \mu_{i-1 \to i}(1) = \exp(-\epsilon \sum_{j=1}^{i-1} \mathcal{L}_r(t_j; \theta)),$$

which is equivalent to the definition of causal weights.

## 4 MESSAGE-PASSING WEIGHTS FOR TRAINING PINNS FOR TIME-DEPENDENT PDES

The remaining problem is how to build the weights adapting the connection in Section 3. In fact, the normalizing process of BP algorithm was not displayed in Section 3, since they are already naturally normalized in L1 sense. Now, we shall define the weights for training PINNs, namely *Message-Passing weights (MP weights)*, motivated by Section 3, which resemble the message-passing process of BP algorithm without the normalization process. For details, we first consider the domain $\Omega = [0, T) \times \prod_{k=1}^{d} I_k \subset \mathbb{R}^{d+1}$ whose boundary consist of the Cartesian products of closed intervals. Next, we define the undirected graph $G = (V, E)$ whose nodes are the set of uniformly spaced collocation points of $\Omega$ with a gap size of $\delta$. Then we can define $G$ as follows:

$$V = \{(t_j, x_{i_1}, \ldots, x_{i_d}) : j = 1, \ldots, N_t, i_k = 1, \ldots, N_k, k = 1, \ldots, d\}$$

$$E = \{(v, w) \in V^2 : ||v, w||_{L^1} = \delta\} \setminus (\{0\} \times \prod_{k=1}^{d} I_k)^2$$

where $|| \cdot ||_{L^1}$ is an L1 norm. Then Motivated from the Eq. 4 and the connection in 3, we can define the message-passing process on this graph, with the customized messages $\mu_{y \to x}^{(\tau)}(\theta) \in [0, 1]$ calculated for internal iteration steps $\tau = 1, \ldots, D$ in each training loop, as:

$$\mu_{y \to x}^{(\tau+1)}(\theta) = \exp(-\epsilon \mathcal{N}[u_\theta](y)^2) \prod_{(y,z) \in E, z \neq x} \mu_{z \to y}^{(\tau)}(\theta), \qquad \forall (x, y) \in E.$$

Especially, for two-dimensional time dependent case with $\Omega = [0, T] \times I$, the messages are defined as:

$$\mu_{(t,x) \to (t-\delta,x)}^{(\tau)}(\theta) = 1,$$

$$\mu_{(t,x) \to (t+\delta,x)}^{(\tau+1)}(\theta) = \exp(-\epsilon \mathcal{N}[u_\theta](t,x)^2) \mu_{(t-\delta,x) \to (t,x)}^{(\tau)}(\theta) \mu_{(t,x-\delta) \to (t,x)}^{(\tau)}(\theta) \mu_{(t,x+\delta) \to (t,x)}^{(\tau)}(\theta),$$

$$\mu_{(t,x) \to (t,x-\delta)}^{(\tau+1)}(\theta) = \exp(-\epsilon \mathcal{N}[u_\theta](t,x)^2) \mu_{(t-\delta,x) \to (t,x)}^{(\tau)}(\theta) \mu_{(t,x-\delta) \to (t,x)}^{(\tau)}(\theta),$$

$$\mu_{(t,x) \to (t,x+\delta)}^{(\tau+1)}(\theta) = \exp(-\epsilon \mathcal{N}[u_\theta](t,x)^2) \mu_{(t-\delta,x) \to (t,x)}^{(\tau)}(\theta) \mu_{(t,x+\delta) \to (t,x)}^{(\tau)}(\theta)$$

whenever $t - \delta, x - \delta$ and $x + \delta$ are valid coordinates of points in $V$ with fixed conditions

$$\mu_{v \to w}^{(\tau)}(\theta) = 1, \qquad \forall v \in V \cap \partial\Omega, \forall w \in V \setminus \Omega$$

for each inernal iteration $\tau = 1, \ldots, D$ and

$$\mu_{v \to w}^{(1)}(\theta) = 1, \qquad \forall v, w \in V.$$

Finally, we define the MP-weights

$$w^{\text{MP}}(x, \theta) := \prod_{(x,y) \in E} \mu_{y \to x}^{(D)}(\theta), \qquad \forall x \in V$$

which is used to define the new residual loss $\mathcal{L}_r^{\text{MP}}(\theta)$ in Eq. 1 as

$$\mathcal{L}_r^{\text{MP}}(\theta) := \sum_{x \in V} w^{\text{MP}}(x; \theta) |\mathcal{R}[u](x)|^2, \qquad \forall x \in V.$$

To facilitate the better understanding, we visualized the evolution of MP weights during the learning process for viscous Burgers equation in the Figure 1.

In practice, the hyperparameter $D$ was set to 20. Additionally, adjusting causality has a crucial impact on the training and validation of PINNs. If the causality is too small, the differences in the residuals have less effect; if it is too large, convergence of the weights during the training may not be achieved. To measure the efficiency and effectiveness of loss balancing ability simultaneously, we applied the following process for determining causality: the causality parameter $\epsilon$ for each Causal PINN and MP-PINN was initially set to a sufficiently large value (practically set to 1000) and decreased exponentially by a factor of $1/10$. This process was stopped and the value of $\epsilon$ was determined when the weights fully converged to 1, i.e. $||w - 1|| < \delta = 0.1$, within the training epochs.

## 5 EXPERIMENTS

To demonstrate the performance of MP-weights, we aggregate the various time-dependent PDE examples as a benchmark from (Krishnapriyan et al., 2021; Wang et al., 2022a; Son et al., 2023), which are Convection, Allen-Cahn, viscous Burgers, and Klein-Gordon equation. Also, to verify the effectiveness of MP-PINN, we compared it with PINN, Causal PINN and AL-PINN that how MP-PINN overcome these methods and analyze the superiority over other adaptive weighting alogrithms. The results are displayed in Table 1. Notably, We aim to show that the superiority of MP-PINN is due to its causal inference capabilities and present the following two experimental settings. First, each PDE has Dirichlet boundary conditions. This is because, from the perspective of supervised learning in PINNs, the initial and boundary conditions serve as true labels, which transmit information to the interior collocation points. Second, we propose early stopping based on loss without any learning scheduler during optimization. Since adaptive weighting algorithms adjust weights according to the residual magnitudes from the governing equation, initial, and boundary conditions, they can effectively guide the learning process. This suggests that these algorithms have an inherent ability to self-regulate learning. To directly compare this capability, we performed full-batch training using the Adam optimizer for 300,000 epochs. Finally, we used uniformly spaced collocation points, as well as initial and boundary points, for the training and test datasets.

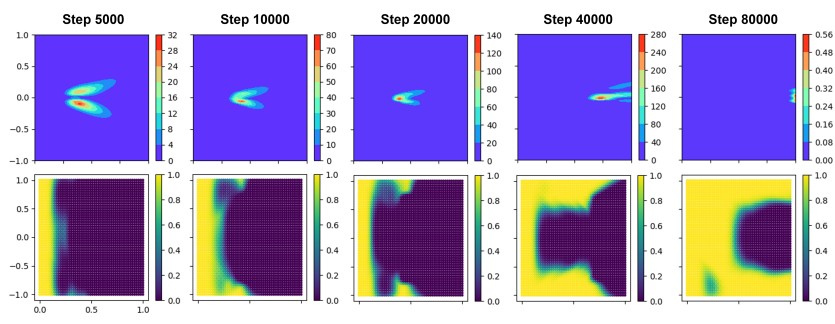

Figure 1: The process of increasing MP weights for viscous Burgers equation displayed with squared residuals (First row) and MP weight value (Second row) for each collocation points.

Table 1: Comparison of Relative L2 errors.

| Experiments | PINNs | Causal PINNs | AL-PINNs | MP-PINNs |
|---|---|---|---|---|
| Convection | 7.91e-03 | **2.34e-03** | 7.93e-03 | 2.51e-03 |
| Allen-Cahn | 5.24e-01 | 6.85e-02 | 6.45e-01 | **4.34e-02** |
| viscous Burgers | 3.18e-01 | 1.47e-02 | 4.29e-01 | **1.16e-02** |
| Klein-Gordon | 4.18e-03 | 2.43e-02 | 3.28e-03 | **1.26e-03** |

## 5.1 CONVECTION EQUATION

We first consider the Convection equation given as

$$
\begin{aligned}
\partial_t u + \beta \partial_x u &= 0, & \text{for } (t, x) \in [0, 1) \times [0, 2\pi], \\
u(0, x) &= \sin(x), & \text{for } x \in [0, 2\pi], \\
u(t, x) &= \sin(-\beta t), & \text{for } (t, x) \in [0, 1) \times \{0, 2\pi\}
\end{aligned}
$$

with $\beta = 30$. Figure 2 illustrates the absolute error between the true solution and the approximated solutions of each PINN. Both Causal PINN and MP-PINN achieved significantly better approximations compared to other PINNs. This demonstrates that MP-PINN has also successfully learned the time dependency in this problem.

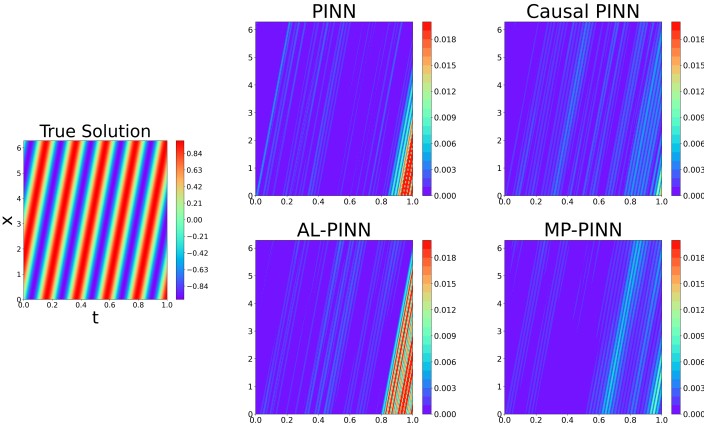

Figure 2: The true solution for Convection equation and comparison of absolute error map $|u_\theta - u|$ for each PINNs. In the error map, all values exceeding $0.02$ were clipped to a constant.

## 5.2 ALLEN-CAHN EQUATION

Next, we consider the Allen-Cahn equation given as

$$\partial_t u + \beta \partial_x^2 u + 5u(u^2 - 1) = 0, \qquad \text{for } (t, x) \in [0, 1) \times [-1, 1],$$
$$u(0, x) = x^2 \cos(\pi x), \qquad \text{for } x \in [-1, 1],$$
$$u(t, x) = -1, \qquad \text{for } (t, x) \in [0, 1) \times \{-1, 1\}$$

with $\beta = 10^{-4}$. Figure 3 illustrates that both Causal PINN and MP-PINN outperform other PINNs. However, this example demonstrates that spatial causality is also important for training PDE. Figure 4 shows the cross-section of the Allen-Cahn equation where the mean absolute error is the largest for each weight across all time grids. As demonstrated, while Causal PINN fails to preserve spatial causality, MP-PINN maintains it even in the worst-case scenario. This highlights that MP-PINN respects the spatial causality of Allen-Cahn equation.

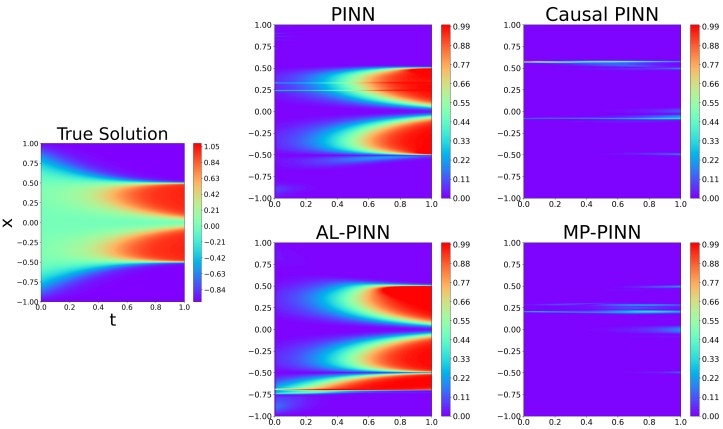

Figure 3: The true solution for Allen-Cahn equation and comparison of absolute error map $|u_\theta - u|$ for each PINNs. In the error map, all values exceeding 1 were clipped to a constant.

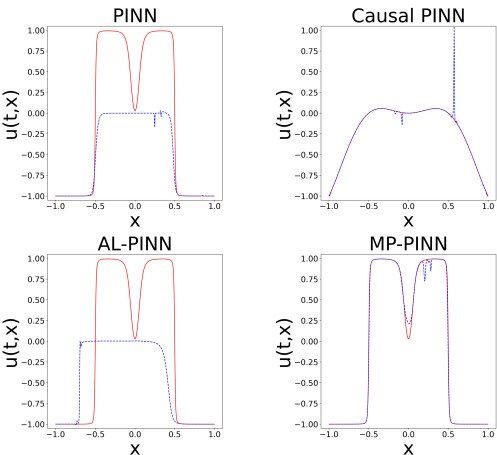

Figure 4: The comparison between the true solution (red dashed line) and the approximated solution (blue dotted line) from each PINN for the Allen-Cahn equation is shown at the time where the mean absolute error of the cross-section is the largest. The selected time points are $t = 0.0$ for Causal PINN and $t = 1.0$ for the others.

## 5.3 VISCOUS BURGERS EQUATION

To emphasize that MP-PINN effectively respects spatial causality, we considered the following viscous Burgers equation:

$$\partial_t u + u \partial_x u - \nu \partial_x^2 u = 0, \qquad \text{for } (t,x) \in [0,1) \times [-1,1],$$
$$u(0,x) = -\sin(\pi x), \qquad \text{for } x \in [-1,1],$$
$$u(t,x) = 1, \qquad \text{for } (t,x) \in [0,1) \times \{-1,1\}$$

where $\nu = 0.01/\pi$. As demonstrated by the true solution in Figure 5, the viscous Burgers equation exhibits very sharp spatial variations along the horizontal line $x = 0$. While Causal PINN struggles to capture these changes, MP-PINN effectively smooths the steep spatial errors.

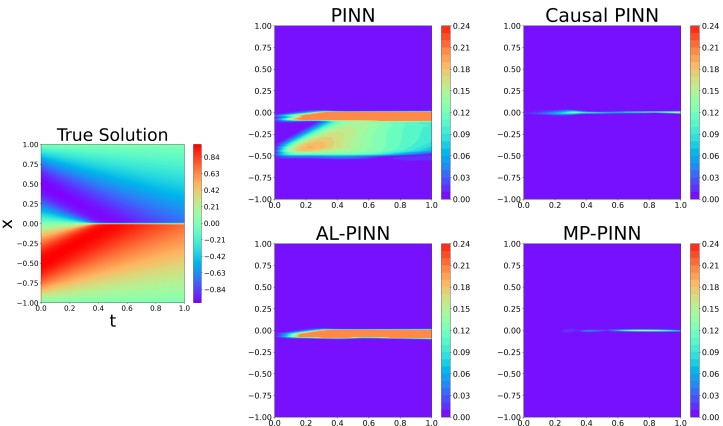

Figure 5: The true solution for viscous Burgers equation and comparison of absolute error map $|u_\theta - u|$ for each PINNs. In the error map, all values exceeding $0.2$ were clipped to a constant.

## 5.4 KLEIN-GORDON EQUATION

Finally, we will investigate whether the MP-PINN effectively learns in the case where the second derivative, rather than the first derivative of time, is provided in the given equation. For this, We consider the Klein-Gordon Equation given as

$$\partial_t^2 u - \partial_x^2 u + u^3 = f(t,x), \qquad \text{for } (t,x) \in [0,1) \times [0,1],$$
$$u(0,x) = g_1(x), \qquad \text{for } x \in [0,1],$$
$$\partial_t u(0,x) = g_2(x), \qquad \text{for } x \in [0,1],$$
$$u(t,x) = h(t,x), \qquad \text{for } (t,x) \in [0,1) \times \{0,1\}.$$

with the unknown $f, g_1, g_2, h$ are derived from the pre-given solution

$$u(t,x) = x\cos(5\pi t) + (tx)^3.$$

Figure 6 illustrated that all benchmark datasets, including Causal PINN, fail to capture the significant information transmitted from the upper boundary $x = 1$, resulting in large errors in the central diagonal region. In contrast, MP-PINN successfully reduces these errors, providing evidence of respecting spatial causality.

## 6 CONCLUSIONS

In this study, we have developed a novel method, MP weights which respect the causality between the training of each collocation points. This was achieved by considering not only time dependency but also spatial causality, leading to superior performance respect to other adaptive weighting algorithms and the ability to handle a wider variety of PDE types.

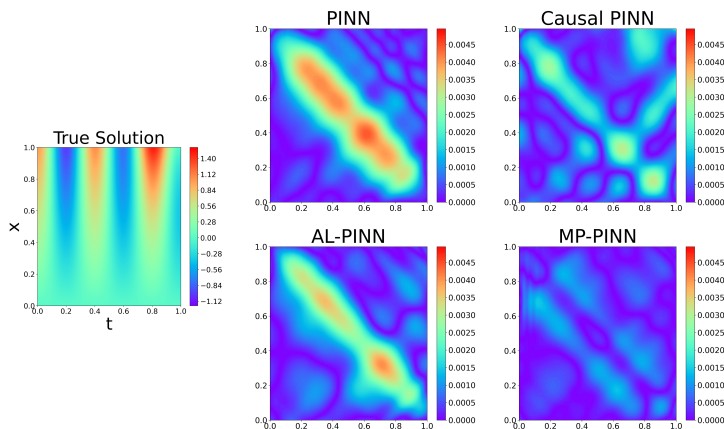

Figure 6: The true solution for Klein-Gordon equation and comparison of absolute error map $|u_\theta - u|$ for each PINNs.

A limitation of our research is the increase in computational cost for weight calculations and running time. Additionally, further theoretical analysis is required for the hyperparameter $D$, which represents the number of message-passing iterations. Specifically, as the number of collocation points increases, $D$ inevitably needs to increase as well, requiring proper adjustment.

While this paper concludes with just a modification of the BP algorithm in the context of MP-PINN, in future research, we plan to investigate whether this can be related to actual causal inference. Additionally, we expect that more experiments for general types of PDEs can be conducted.

## 7 REPRODUCIBILITY STATEMENT

We provide the detailed experimental setup for each PDEs and PINNs in Appendix A.

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

## A    EXPERIMENTAL SETUP

We adopt the true solution as the analytic solution for Convection, viscous Burgers and Klein-Gordon equation. For Allen-Cahn equation, we use chebfun package Platte & Trefethen (2010) to generate the numerical solution. The training settings for the benchmark PINNs, except for those specifically mentioned in the main text, are consistent with those in (Krishnapriyan et al., 2021; Wang et al., 2022a; Son et al., 2023), as detailed in Table 2.

Table 2: Settings for training are displayed for each PINN (VN: PINN, CS: Causal PINN, AL: AL-PINN and MP: MP-PINN). For the network structure, MLP refers to a fully-connected network with hyperbolic tangent as activation functions, where all layers are initialized via Xavier initialization, and ResNet refers to an MLP with residual connections. The numeric values for Network Structure denote (hidden layer width) $\times$ (hidden layer depth). The remaining settings are represented as follows: learning rate $\eta_\theta$ for PINNs, initial and boundary coefficients $\lambda_{ic}$ and $\lambda_{bc}$ respectively, causality parameter $\epsilon$ for CS and MP, and learning rate $\eta_\lambda$ for weight of AL.

| Experiment | Weight | Network Structure | Train Grid / Test Grid | $\eta_\theta$ | $\lambda_{ic}, \lambda_{bc}$ | $\epsilon$ | $\eta_\lambda$ |
|---|---|---|---|---|---|---|---|
| Convection | VN | MLP, $50 \times 4$ | $50 \times 50$ / $201 \times 512$ | $10^{-3}$ | $10^2$ | - | - |
| | CS | | | | | $1$ | - |
| | AL | | | | | - | $1$ |
| | MP | | | | | $10^{-5}$ | - |
| Allen-Cahn | VN | MLP, $128 \times 4$ | $50 \times 50$ / $201 \times 512$ | $10^{-3}$ | $10^2$ | - | - |
| | CS | | | | | $10^2$ | - |
| | AL | | | | | - | $1$ |
| | MP | | | | | $10^{-4}$ | - |
| viscous Burgers | VN | ResNet, $64 \times 8$ | $50 \times 50$ / $100 \times 200$ | $10^{-4}$ | $1$ | - | - |
| | CS | | | | | $10^2$ | - |
| | AL | | | | | - | $10^{-3}$ |
| | MP | | | | | $10^{-4}$ | - |
| Klein-Gordon | VN | ResNet, $64 \times 8$ | $50 \times 50$ / $100 \times 200$ | $10^{-3}$ | $5 \times 10^2$ | - | - |
| | CS | | | | | $10^{-2}$ | - |
| | AL | | | | | - | $1$ |
| | MP | | | | | $10^{-8}$ | - |

## B  ADDITIONAL COMPARISON OF RESPECTING SPATIAL CAUSALITY

To clarify the results, we provide figures Figure 7, 8 and 9, comparing the cross-sectional error for other equations except Allen-Cahn.

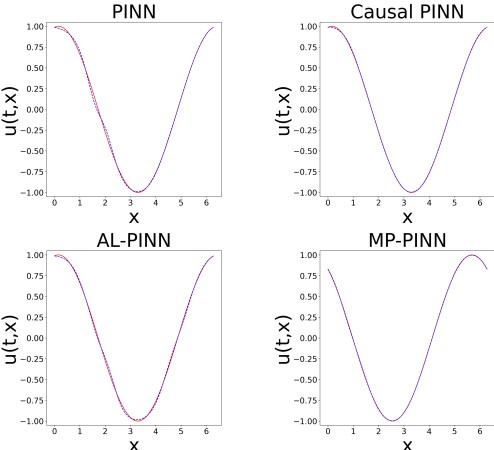

Figure 7: The comparison between the true solution (red dashed line) and the approximated solution (blue dotted line) from each PINN for the Convection equation is shown at the time where the mean absolute error of the cross-section is largest. The selected time points are $t = 0.98$ for MP-PINN and $t = 1.0$ for the others.

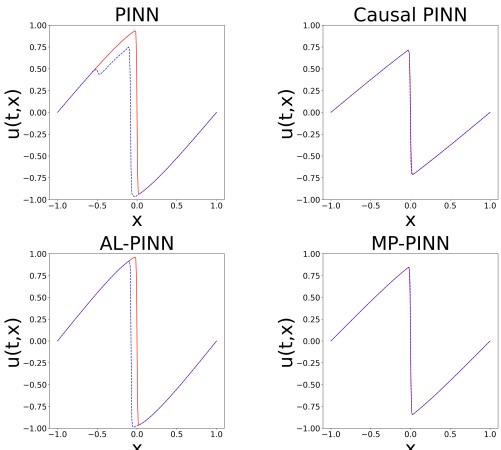

Figure 8: The comparison between the true solution (red dashed line) and the approximated solution (blue dotted line) from each PINN for the viscous Burgers equation is shown at the time where the mean absolute error of the cross-section is largest. The selected time points are $t = 0.62$, $t = 1.0$, $t = 0.57$ and $t = 0.77$ for PINN, Causal PINN, AL-PINN and MP-PINN, respectively

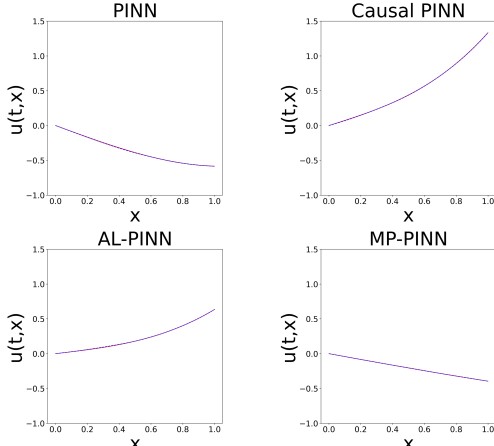

Figure 9: The comparison between the true solution (red dashed line) and the approximated solution (blue dotted line) from each PINN for the Klein-Gordon equation is shown at the time where the mean absolute error of the cross-section is largest. The selected time points are are $t = 0.64$, $t = 0.85$, $t = 0.72$ and $t = 0.27$ for PINN, Causal PINN, AL-PINN and MP-PINN, respectively

