# OpenReview forum: "Physics-Informed Neural Networks with Message-Passing Weights"
_ICLR.cc/2025/Conference — ICLR 2025 Conference Withdrawn Submission_

### Official Review · Reviewer_vW7j · 2024-10-18

**Soundness:** 3
**Presentation:** 3
**Contribution:** 3
**Rating:** 6
**Confidence:** 2

**Summary:**

This paper analyzes the setting of Causal PINNs, which reformulates the standard PINN loss function to incorporate information about time dependencies into weights. The authors provide theoretical justification for Causal PINNs by analyzing the belief propagation algorithm. Specifically, they find that running BP over a Markov random field yields the weighting used in Causal PINNs. Next, the authors propose a message passing-based alternative form of PINNs that outperforms adaptive weighting methods for PDEs. Empirically, they demonstrate that these perform better than alternative PINN approaches on a variety of time-dependent PDEs.

**Strengths:**

1. The paper provides a new theoretical understanding of Causal PINNs.
2. Experiments show that MP-PINNs outperform existing algorithms in handling time-dependent PDEs, especially with respect to spatial causality.

**Weaknesses:**

1. The authors mention that a limitation of their paper is the lack of theoretical analysis on the hyperparameter D (the number of iterations of message passing). While I believe that theoretical analysis may not be necessary, I think that ablation studies could be run to empirically understand the impact of this hyperparameter on performance.

**Questions:**

How are the L2 errors in Table 1 computed? Are these averaged over time t?

---

### Official Review · Reviewer_8bpG · 2024-11-03

**Soundness:** 2
**Presentation:** 1
**Contribution:** 2
**Rating:** 3
**Confidence:** 2

**Summary:**

This paper presents a message-passing algorithm for solving partial differential equations (PDEs). The main idea seems to be to discretize the input space and use the loss function of Physics-Informed Neural Networks (PINNs) for an exponential error propagation across time and space. Results for four different kind of PDEs are presented and show good results using the message passing scheme.

**Strengths:**

The paper presents an alternative (iterative) algorithm for solving PDEs where the initial- and boundary-conditions serve as training points.

**Weaknesses:**

I am not an expert in PDEs and found Section 2.1 incomprehensible: For example, isn't $\partial_t u$ already a linear differential operator (first derivative). What is the role of $\mathcal{R}$? Also, I was not able to follow the presentation of the main idea in Section 3; how does $v_i$ and resolve to $x$ and $t$ in Section 2? The experimental results seem convincing but I am not remotely able to understand the algorithmic idea that the authors are using. I would urge the authors to significantly improve the accessibility of their idea and algorithm.

**Questions:**

* Line 108: It seems that the font of $N$ and $B$ differs from that in line 102 (where $\mathcal{N}$ and $\mathcal{B}$ is used)
* Line 139: Shouldn't this be $\propto$ rather than $=$?
* Line 156: G should be $G$
* Line 168: There is an extra comma after $N_t$

---

### Official Review · Reviewer_Hv5w · 2024-11-04

**Soundness:** 3
**Presentation:** 1
**Contribution:** 3
**Rating:** 3
**Confidence:** 4

**Summary:**

The paper attempts to provide a theoretical analysis of the effectiveness of Causal PINNs with the Belief Propagation algorithm and proposes a new algorithm called Message Passing PINNs. It demonstrates the effectiveness of this approach in solving various PDEs.

**Strengths:**

The empirical performance looks promising.

**Weaknesses:**

The paper does not seem ready for publication; the clarity is lacking, and parts of the related work and formal justification for the theoretical analysis are missing.

1. **Missing related work**: In line 80, “Despite the applicability of PINNs for solving various PDEs, there are still various problems to solve.” However, the paper does not cite any work here and later cites only one paper afterward. In line 90, “to address the failure mode of PINNs.” The paper does not describe what the failure mode of this algorithm is.

2. **Insufficient background**: The paper has not defined the main goal or the notation used in section 2, which may make it difficult to read.

3. **Theoretical analysis organization**: The paper does not present the results as theorems and proofs but rather writes the results in paragraph form, making it less formal.

While the empirical performance looks good, I think the current version of this paper is not yet ready for ICLR.

**Questions:**

see weakness

---

### Official Review · Reviewer_einq · 2024-11-07

**Soundness:** 3
**Presentation:** 3
**Contribution:** 2
**Rating:** 5
**Confidence:** 3

**Summary:**

- Provides a theoretical understanding for Causal PINNs through the Belief Propagation algorithm.
- Proposes a novel adaptive weighting algorithm termed Message Passing PINNs (MP-PINNs)
- Empirically demonstrate that the proposed approach MP-PINNs outperform existing adaptive weighting methods for solving complex PDEs

**Strengths:**

- Empirically demonstrates the proposed approach MP-PINNs outperform existing adaptive weighting methods for solving complex PDEs
- Provides a novel theoretical understanding of Causal PINNs

**Weaknesses:**

- The authors note that further theoretical analysis is required for the hyper parameter D (number of message-passing iterations). A proper ablation study to empirically understand performance w.r.t. this hyper parameter would be useful.

**Questions:**

- The authors note a limitation of their method is the increase in computational cost for weight calculations; what is the specific computational costs required for this method relative to existing baselines?

---

### Author Response · Authors · 2024-11-21

Dear reviewers,

Thank you for your valuable review of our paper. After careful consideration, we think the current version is inappropriate for ICLR, and we have decided to withdraw our paper to make further improvements.

---

### Note · Authors · 2024-11-21

I have read and agree with the venue's withdrawal policy on behalf of myself and my co-authors.